Spatio-temporal modeling of co-dynamics of smallpox, measles, and pertussis in pre-healthcare Finland

Pasanen Tiia-Maria tiia-maria.h.pasanen@jyu.fi 1
Helske Jouni 1 2
Högmander Harri 1
Ketola Tarmo 3 4
1 Department of Mathematics and Statistics, University of Jyväskylä , Jyväskylä , Finland
2 INVEST Research Flagship Centre, University of Turku , Turku , Finland
3 Department of Forestry, University of Helsinki , Helsinki , Finland
4 Department of Biological and Environmental Science, University of Jyväskylä , Jyväskylä , Finland
Geard Nicholas
Electronic publication date: 2024 Sep 26
Publication date: 2024
Volume: 12
Electronic Location ID: e18155
Received 2024 Feb 6; Accepted 2024 Sep 1
Copyright: ©2024 Pasanen et al.
Copyright year: 2024
Copyright holder: Pasanen et al.
License: This is an open access article distributed under the terms of the Creative Commons Attribution License, which permits unrestricted use, distribution, reproduction and adaptation in any medium and for any purpose provided that it is properly attributed. For attribution, the original author(s), title, publication source (PeerJ) and either DOI or URL of the article must be cited.
License URL: https://creativecommons.org/licenses/by/4.0/

Keywords: Spatio-temporal, Infection co-dynamics, Pertussis, Measles, Smallpox, Bayesian analysis

Funding: The Finnish Cultural Foundation, the Emil Aaltonen Foundation and the Research Council of Finland 331817 The Research Council of Finland 331817 355153 The Research Council of Finland 278751 Tiia-Maria Pasanen was supported by the Finnish Cultural Foundation, the Emil Aaltonen Foundation and the Research Council of Finland grant 331817. Jouni Helske was supported by the Research Council of Finland grants 331817 and 355153. Tarmo Ketola was supported by the Research Council of Finland grant 278751. The funders had no role in study design, data collection and analysis, decision to publish, or preparation of the manuscript.

==============================
Infections are known to interact as previous infections may have an effect on risk of succumbing to a new infection. The co-dynamics can be mediated by immunosuppression or modulation, shared environmental or climatic drivers, or competition for susceptible hosts. Research and statistical methods in epidemiology often concentrate on large pooled datasets, or high quality data from cities, leaving rural areas underrepresented in literature. Data considering rural populations are typically sparse and scarce, especially in the case of historical data sources, which may introduce considerable methodological challenges. In order to overcome many obstacles due to such data, we present a general Bayesian spatio-temporal model for disease co-dynamics. Applying the proposed model on historical (1820–1850) Finnish parish register data, we study the spread of infectious diseases in pre-healthcare Finland. We observe that measles, pertussis, and smallpox exhibit positively correlated dynamics, which could be attributed to immunosuppressive effects or, for example, the general weakening of the population due to recurring infections or poor nutritional conditions.

Introduction

Infections exist rarely in isolation, and their effects on hosts are known to interact and to have both positive and negative relationships between each other (Gupta, Ferguson & Anderson, 1998; Rohani et al., 2003; Shrestha et al., 2013; Mina et al., 2015; Nickbakhsh et al., 2019). For example, cross immunity may prevent others from infecting the host, competition for the same resources or susceptible host can have strong effects on epidemics, and sometimes one infection paves a way for another (Gupta, Ferguson & Anderson, 1998; Rohani et al., 2003; Graham, 2008). Perhaps historically the best-known relationship between infections is the immunosuppressive effect of measles on the following pertussis epidemic by increasing the severity of the epidemic (see Coleman, 2015; Mina et al., 2015; Noori & Rohani, 2019). Coinfections of parasites (Graham, 2008) and viruses and respiratory bacterial infections are well known (e.g., Bakaletz, 2017; Wong et al., 2023), whereas understanding coinfections and cotransmissions of, for example, zika, dengue, and chikungunya viruses presents a current serious challenge for public health (Vogels et al., 2019).

Demographic consequences of epidemics are most dramatically seen in large cities and in densely populated areas, which is reflected in the epidemiological research in general (Mueller et al., 2020). However, as rural areas constitute a large part of most countries, the spatio-temporal dynamics of epidemics in populations with low densities deserve more attention (Mueller et al., 2020). In rural areas, populations often consist of loosely connected metapopulations rather than large and epidemiologically more autonomous populations in cities. This has most likely strong repercussions for the drivers of epidemics (Ball et al., 2015) and also for the co-occurrence of infections. However, these issues are rarely addressed in the literature, possibly due to the statistical challenges encountered with sparse and scarce data, as well as the difficulty of modeling the dynamics of several infections simultaneously both in space and time.

The discrepancy between studying dense and sparse populations is evident and can be seen, for example, by comparing our case of rural Finland in 1820–1850 to the seminal research of Rohani et al. (2003). Their study is based on five large European cities, where the weekly number of fatalities frequently exceeds 30 and even 80. In our data, the recorded incidents in most of the towns rarely exceed one person per month, as we study a small and mainly agrarian population in the southern part of Finland with circa 1.2 to 1.6 million individuals (Voutilainen, Helske & Högmander, 2020). The population, without proper healthcare (Saarivirta, Consoli & Dhondt, 2012), was spread over a vast area in geographically separated, but socially connected, small towns and villages. Based on the data from 1882, population sizes of towns varied between 300 and 25, 000 (Statistical Office of Finland, 1882; Ketola et al., 2021). Despite the obvious uncertainty of population censuses during that era (Voutilainen, Helske & Högmander, 2020), the contrast between our data and most of the published datasets is striking. Statistical modeling of such data is problematic due to incomplete information from some locations and the rare occurrence of events, hampering the ability of generally used models to consider several infectious diseases at the same time and on both temporal and spatial scales.

To estimate the spatio-temporal co-dynamics of deaths due to pertussis, measles, and smallpox, we build a model that can overcome the limitations inherent in our data. The model jointly estimates the spread of multiple infections, enabling the exploration of the temporal and spatial dependence structures both within and between the infections. Our general Bayesian model consists of a multivariate latent incidence process, a seasonal component, and multiple predictors whose effects may vary between the towns. This allows us to study the dynamics of the diseases simultaneously despite having only incomplete information about the deaths. The results we get from modeling the mere presence-absence data are compared with those of modeling the corresponding death counts, and the simplification is deemed to be a reasonable option in our case. Given the limitations of our data, we do not aim to make causal claims on a biological level, and rather than focusing on the magnitude of infections and the intensity of deaths, our primary interest lies in understanding how the prevalence of these three diseases varied both spatially and temporally in pre-industrial Finland, and if there were possible associations in these dynamics across the diseases.1

Materials and methods

Data

During the study period, 1820–1850, the parishes in Finland kept track, among others, of baptisms, burials and causes of deaths, according to common and long held principles (Pitkänen, 1977). Even though the death diagnostics were based on symptoms, some infections can be considered to be diagnosed rather accurately due to their characteristic features. These diseases include pertussis (whooping cough), measles, and smallpox, which we consider. These infections were the main reason for child mortality, and, overall, they were responsible for approximately 5, 3, and 3 percent of total deaths, respectively, according to our data. Based also on the available records in our data, the median ages of deaths in complete years were 0 (sd = 3.6) for pertussis, 2 (sd = 3.9) for measles, and 2 (sd = 7.5) years for smallpox. Smallpox vaccinations were started in Finland in 1802 and were slowly progressing during the study period (Briga, Ketola & Lummaa, 2022). However, general healthcare was almost non-existent as in 1820 there were only 373 hospital sickbeds for 1.2 million inhabitants (Saarivirta, Consoli & Dhondt, 2012).

Our data consist of the daily numbers of deaths, classified by the cause of death, between January 1820 and December 1850 from N = 387 different regions (towns) in mainland Finland with the exclusion of northern areas. The time window is chosen such that there were no major famines, wars, border changes, or other potentially confounding events, which could have altered the geographical partition or the dynamics of the epidemics. The general stability achieved is advantageous in the modeling.

Although using a daily time scale would, theoretically, be ideal for modeling disease dynamics, the infrequency of deaths implies that our data lack sufficient information to study temporal dependencies at such a detailed level. This issue is likely further pronounced by the spatial heterogeneity of the data and the potentially complex lagged auto- and cross-dependencies between the infection dynamics of the diseases. Therefore, the daily counts of deaths are aggregated over time into a monthly level, decreasing the number of zero observations yet maintaining a reasonable time resolution for observing the spread of the diseases on our geographical scale. This yields a total of T = 372 time points. The counts of the observed numbers of deaths by disease, considering the aggregated data, are visualized in Fig. 1. In each case of the three infections, about 93–96% of the death counts are zero or one, and less than 2% of the counts are more than three, despite the aggregation. The reliability of the actual death counts varies considerably both temporally and spatially owing to the heterogeneous quality of the parish records and the cause of death classifications. Moreover, these deficiencies are not necessarily independent of the true number of deaths. It is also noteworthy that despite the records of baptisms and burials, there are no reliable estimates of the population sizes at the town level (Voutilainen, Helske & Högmander, 2020). Hence the intuitive idea of using local relative mortality is unfortunately beyond reach. Because of this and the aforementioned reliability issues—and since most of the counts are still either zero or one even after the aggregation to a monthly level—only the dichotomous knowledge of the death occurrence is used in the main analysis. Even so, the count data are considered in the model comparison section.

Figure 1 The counts of the observed numbers of deaths over all towns and months plotted by disease.

The first bar indicates the number of missing observations. The numbers above the bars are the counts. Note that the vertical axis is on a logarithmic scale.

About 24% of the data concerning death occurrences in a town and a month are missing, and from 57 out of the overall 387 regions there are no observations at all. The missingness pattern is common to all the diseases, as for a particular town and month, we have observed the death counts for all the diseases or for none of them. This is because the missing data can be attributed to the absence of parish records that document all deaths. Such unavailability of data may be a result of incomplete digitization of the parish records, loss of the documents for example due to a fire in the local rectory or church, or simply because there were no deaths in a given month. Therefore, the missing data could potentially depend on the unknown regional population size, but not on the specific cause of death, given that the proportions of deaths attributable to the particular cause of death are relatively small. Nevertheless, our proposed model provides estimates of the monthly probabilities of observing at least one death also for the towns with missing observations.

Model

We construct a general model to describe the spatial and temporal dependencies both within and between the infections under study. We also want to enable exploiting other relevant information as explanatory variables. In epidemiological context, there typically occur spatial or temporal trends or seasonal effects, which can be included as separate components in the model. Due to the nature of our data, we model the probability of observing at least one death caused by a disease in a certain town in a certain month. The model consists of a trend, a seasonal effect, and a regression part reflecting the local effects of the previous state of infection in the focal town and its neighboring towns.

Formally, let yi,td denote a dichotomous variable of an event where at least one death occurs due to a disease d in a region i = 1, …, N at a time point t = 1, …, T, where N is the number of regions and T the number of time points. Let Kxd indicate the number of explanatory variables x, based on the features of the region i. Accordingly, z denotes the explanatory variables, and Kzd their number, related to the neighborhood of the region i. Thus the model for observing at least one death caused by the disease d in the region i at the time point t can be written as follows: (1) yi,td∼Bernoullilogit−1ηi,td,

where (2) ηi,td=λidτtd+std+aid+bid ∑k=1Kxdβkdxi,kd+cid ∑k=1Kzdγkdzi,kd.

Here Bernoulli distribution with a logit link is chosen due to our dichotomous consideration of the occurrences of death, but other distributions with appropriate link functions can be applied for different types of response variables.

The first three terms being summed in Eq. (2) form a base level, in our case, for the probability of observing at least one death caused by the disease d at each town and month. More specifically, the first term consists of the time dependent latent factor τt, describing the nationwide incidence (on log-odds scale), or trend, of the disease d, and the regional adjustments or loadings λi, with respect to the mean level. As in general dynamic factor models, the products λiτt are not identifiable without constraints (Bai & Wang, 2015). Instead of the common approach of fixing one of the loadings λi to 1, we constrain the mean of the loadings to 1, enabling the interpretation of the factor τt as the nationwide incidence level. Due to the nature of the other terms in Eq. (2), this incidence level gives the national average log-odds of observing at least one death in an “average” town in a given month if no deaths were observed in the previous month in the focal town or in its neighboring towns. The second term st is a monthly seasonal effect, which is the average deviation from the nationwide incidence level, summing up to 0 over the months. The third term ai is a regional, zero-mean constant reflecting local deviations from the nationwide incidence level τt due to unobserved local demographic, geographic, social or other characteristics associated with mortality.

The last two sum terms in Eq. (2) form the regression part of the model. The first sum includes the covariates xi,k related to the focal region i, and the second sum the covariates zi,k related to the neighboring regions. These variables have both nationwide coefficients β and γ, and their local adjustments bi and ci amplifying or diminishing the nationwide level. As the regional constants ai, also the multiplicative local adjustments may account for any unobserved heterogeneity between the regions, such as the local population sizes or densities. The adjustment parameter bi reflects the features of the focal region i, and ci those of the neighborhood of the region i (possibly relative to i). We assume that bi is the same for all covariates xk, and, accordingly, ci for all zk, since the underlying regional characteristics modifying the nationwide mortality effects β and γ do not depend on the covariates.

In our study, there are three covariates assigned to the town i and another three to its neighbors for all diseases d = p, m, s, where p stands for pertussis, m measles, and s smallpox. The local explanatory variables xi,k are the presences of deaths caused by the three different diseases in the previous month in the focal town, whereas the regional neighborhood predictors zi,k are the averages of the same presences of deaths over the local neighborhood. We define two regions being neighbors when they share a border. By this definition, all the towns have at least one neighbor. Other definitions of neighborhood could be used as well, for example based on the transportation network or distance, leading to weighted averages of death occurrences.

As noted earlier, our data contain a large number of missing observations. We assume that the probability of having a missing response or predictor variable is independent of the value of the response variable. This presumption may be considered valid, given that the lack of parish records on deaths is unlikely to depend on the causes of deaths in a particular month. Under this assumption, we can model the observed data analogously to a complete-case analysis in an unbiased manner, assuming our model is correctly specified (van Buuren, 2018). This eliminates the need for multiple imputation or sampling missing observations using MCMC algorithms which do not use gradient information (i.e., algorithms capable of sampling discrete variables), both of which would be computationally unfeasible in our Bayesian spatio-temporal context.

In practice, the complete-case analysis in our context means that to use an observation as a response, we require that both the current response variable as well as all of the related covariates are observed. If any of them is missing, we omit the particular combination of town and month as a response. On the other hand, when calculating the neighborhood covariates zi,k, which in our case are the averages over the observations within the neighborhood, we omit the neighbors with missing observations so that they are not included even as a denominator in the evaluation of the mean. If all neighbor observations are missing, the corresponding covariate is defined as missing.

We model the latent factor τt=τtp,τtm,τts, the temporal process describing the baseline of the nationwide incidence rates, as an intercorrelated random walk, τt+1 ∼ N(τt, Σ). Here Σ is an unconstrained 3 × 3 covariance matrix parametrized using the standard deviations στd and the correlation matrix R. In our application this latent process, together with the regional constant and the monthly effect, can be interpreted as the probability to observe at least one new death in a particular town when no deaths caused by any of the three diseases were observed in the previous month in the focal town or in its neighborhood.

In the Bayesian modeling framework, we need prior distributions for all the parameters to be estimated. The incidence factors τd follow a N(−2, 22) prior at the first time point and form a random walk at the later time points. For the correlation matrix R we use an LKJ(1) prior with a Cholesky parametrization (Lewandowski, Kurowicka & Joe, 2009), i.e., a uniform prior over valid 3 × 3 correlation matrices. For the regional parameters we apply Gaussian priors: λid∼N1,σλd2,aid∼N0,σad2,bid∼N1,σbd2, and cid∼N1,σcd2. The prior means of the local adjustments bid and cid are set to 1, which enables us to interpret βkd and γkd as nationwide effects as is the case with additive multilevel models having population-level and group-level effects. While we use a hard equality constraint for the mean of the λids to ensure the identifiability and more efficient estimation of the model, the hierarchical priors for bid and cid are sufficient for their identifiability. For the unknown deviations στd, σλd, σad, σbd, and σcd we assign Gamma priors with shape parameter 2 and rate parameter 1. The nationwide coefficients βkd and γkd have N(0, 22) priors. The seasonal effects std follow a standard normal prior with the aforementioned sum-to-zero constraint. The priors can be seen as weakly informative, and they are chosen primarily to enhance the computational efficiency (Banner, Irvine & Rodhouse, 2020).

Naturally, any of the components in the model could be excluded by setting the corresponding coefficients or standard deviations to zero. Our Bayesian model encompasses all such simplified alternatives, with the corresponding model and parameter uncertainty reflected by the estimated posterior distributions, leading to more truthful uncertainty estimates compared to merely imposing prior constraints on certain effects to be zero.

Results

The model is estimated using Markov chain Monte Carlo (MCMC) with cmdstanr (Gabry & Češnovar, 2022), which is an R interface (R Core Team, 2022) for the probabilistic programming language Stan for statistical inference (Stan Development Team, 2022). To draw the posterior samples we use NUTS sampler (Hoffman & Gelman, 2014; Betancourt, 2018) with four chains, each consisting of 7, 500 iterations, the first 2, 500 of which discarded as a warm-up. With parallel chains the computation takes about ten hours. The model is estimated on a supercomputer node with four cores of Xeon Gold 6230 2.1 GHz processors and 40 GB of RAM. According to the MCMC diagnostics of cmdstanr (Vehtari et al., 2021), the model converges without divergences, the R ^ statistics are always below 1.005, and the effective sample sizes are approximately between 700 and 43, 000. The lowest effective sample size is the one of the deviation parameter of the constants considering measles, σαm. The R and Stan codes, the data used for the analysis, and Supplementary Figures and Tables are available on GitHub (https://github.com/tihepasa/infectionDynamics). All the figures were created using the R packages ggplot2 (Wickham, 2016) and ggpubr (Kassambara, 2023).

To visualize the temporal and spatial patterns of the death occurrences and to see how the model estimates the corresponding probabilities to observe at least one death, the data and the predictions based on the model are plotted as time series and as maps in Figs. 2 and 3. The estimates are computed as ∫pyt+k′|y1,…,yt,θpθ|y1,…,yTdθ. In other words, they are k-step ahead in-sample predictions (often called fitted values in the time series literature, see, e.g., Hyndman & Athanasopoulos (2021)), where k − 1 is the number of preceding missing months, and they are calculated conditional on the posterior distribution of the model parameters, including the latent incidence process τ. For the first time point in this computation we also assume that the missing observations are zeros in order to have covariates for all the sites.

Figure 2 The prevalence of the diseases is illustrated with the dark lines depicting the proportions of the towns where at least one death was observed.

The lighter turquoise lines show the posterior means of the corresponding estimates and the shaded areas their 95% posterior intervals. Note that the data line is calculated over the towns with observations, whereas the estimate line averages all the towns.

Figure 3 The left panels show the proportions of the months when deaths were recorded over the study period of 31 years.

The gray areas indicate the towns where all data are missing. The right panels present the regional averages of the predicted conditional probabilities to observe at least one death caused by each disease in each month given the actual observations from the previous month. Note that the data are averaged over the observed towns and months, whereas the model covers all the towns and months.

The temporal behavior of the data is similar to the corresponding estimates. The slight differences may be due to the fact that the proportions are based only on the data available, whereas the model predictions cover all the month-town combinations. The spatial patterns of the modeled probabilities reflect the infection distributions visible in the data. Pertussis, measles, and smallpox all have emphasis on the eastern half of Finland, with especially measles extending its prevalence to the southern parts of the country as well. When it comes to the completely missing sites, the medians of the estimated average probabilities over time are 3.7 percentage points higher for them than for the sites with observed data in the case of pertussis, 0.3 percentage points higher in the case of measles, and 0.2 percentage points lower in the case of smallpox. The differences are quite small, and by the prediction account, the model seems to work well.

In what follows, we present the results in detail. They confirm that all the components in the model are relevant, capturing different aspects of the spatio-temporal dynamics of the epidemiological phenomena.

The nationwide incidence time series of the diseases are depicted by the factors τtd. The corresponding estimates are shown in Fig. 4 on a probability scale (logit−1τtd). In general, they seem to have the same shapes as the observed nationwide monthly proportions of towns where at least one death caused by pertussis, measles, or smallpox was recorded (see Fig. 2). There is one major disease outbreak regarding smallpox, whereas the other diseases have several peaks, pertussis varying the most. No clear periodicity can be seen in any of the series, which was also confirmed by estimating dominant frequencies via spectral analysis using the R package forecast (Hyndman & Khandakar, 2008).

Figure 4 Posterior means and 95% posterior intervals for the unobserved incidence factors τtd for pertussis, measles, and smallpox over the time period under study.

The curves are on a probability scale.

The seasonal effects std, or the average monthly deviations from the nationwide incidence level, are shown in Fig. 5. According to the estimates, the seasonal effect of pertussis peaks at the beginning of the calendar year, while the effect decreases during the summer and increases again towards the end of the year. In contrast, the only distinctive seasonal effects related to measles and smallpox are the peaks in the spring and the minor decreases at the end of the year.

Figure 5 Posterior means (black), 50% (dark turquoise) and 95% (light turquoise) posterior intervals of monthly seasonal effects std for pertussis, measles, and smallpox over a year.

Measured by the τ factors, we found a distinctive correlation between the infections of measles and pertussis, 0.33 with a 95% posterior interval [0.08, 0.55]. Omitting the specific seasonal term s in the model yields almost the same correlation 0.31 [0.10, 0.50]. The correlation between smallpox and measles is ambiguous, being 0.24 [ − 0.01, 0.46], though it increases to 0.46 [0.26, 0.63] in the model without the seasonal component. This implies that monthly effects explain partly but not exhaustively the connection between these diseases. Smallpox and pertussis seem to be mutually independent, 0.06 [ − 0.19, 0.30], which is also the case with the model without the seasonal terms, 0.15 [ − 0.07, 0.37].

According to the regional loadings λid, adjusting the nationwide factors τtd, it was more likely to die of any of these diseases in eastern and southeastern Finland than in other parts of the study area. This is also in accordance with the maps of the data in Fig. 3. The posterior means of the loadings λ are plotted in Fig. 6. Considering the loadings, there is most local variation in pertussis, σλp = 0.35 with a 95% posterior interval [0.31, 0.40]. With regard to measles and smallpox, the loadings vary less, σλm = 0.17 [0.14, 0.20] and σλs = 0.15 [0.13, 0.17].

Figure 6 The left panels show the posterior means of the local loadings λi, adjusting the national incidence factors τt.

Since the factors are negative, the smaller the loading is, the greater the probability of at least one death is. The right panels illustrate the posterior means of the regional constants ai.

The final term affecting the base level of the probability to observe at least one death caused by pertussis, measles, or smallpox consists of the regional constants ai, shown in Fig. 6. Those related to pertussis and smallpox seem to be larger in eastern and southwestern inland areas, whereas those considering measles are the largest in southern Finland.

The estimates of the nationwide regression coefficients β and γ are represented in Table 1. All effects differing from zero are positive, meaning they increase the probability to detect at least one death. The probability to observe one or more deaths induced by one of these diseases is increased most prominently if there are recorded deaths caused by the same disease in the same town, or in its neighbors, in the previous month. However, there is more uncertainty in the effects of neighbors than in those of the towns themselves. The risk that there is at least one death caused by pertussis is increased by the occurrence of measles, whereas the corresponding effect of smallpox is not distinctive. Measles is probably affected more by smallpox than by pertussis. In turn, measles seems to affect smallpox more than pertussis does.

When it comes to the local adjustments bi and ci, their standard deviations are clearly above zero, varying between 0.22 (σbm) and 0.55 (σcp), which indicates that the local adjustments differ geographically. There are no obvious interpretations of their spatial patterns (see maps of b and c in Supplementary Figures 1 and 2 on GitHub). This is credible since the coefficients represent the combined effects of multiple unobserved features that are not necessarily spatially organized.

For full results of all time and town invariant parameter estimates with their prior and posterior intervals, see Supplementary Table 1 on GitHub.

Model comparison

While our main interest was studying the past spatio-temporal dynamics of infections and disease associations within and between the diseases, we also examined the necessity and reasonableness of modeling the disease interdependencies and the response aggregation. We compared our model with a corresponding one without the dependencies between the infections by excluding the other diseases as explanatory variables and omitting the correlation between the incidence factors τt in the simpler model. Since the original data contained the numbers of deaths instead of the dichotomous aggregates we used as a response, we also estimated corresponding models with the difference of using the counts as a response and a negative binomial distribution to model them. Additionally, the briefly aforementioned model without a seasonal component was included in the comparison in the case of both types of responses. This resulted in six different model versions for comparison: dependent diseases, independent diseases, and dependent diseases without a seasonal effect, each for both Bernoulli and negative binomial distributions.

Table 1 Posterior means and 95% posterior intervals of the nationwide regression parameters grouped by the response disease.

The superscript indicates the response disease and the subscript the explanatory disease.

	Within towns	Between towns	
		Mean	(2.5, 97.5%)		Mean	(2.5, 97.5%)	
pertussis → pertussis	βpp	1.56	(1.47, 1.64)	γpp	1.23	(1.10, 1.35)	
measles → measles	βmm	1.90	(1.82, 1.98)	γmm	2.21	(2.06, 2.38)	
smallpox → smallpox	βss	2.43	(2.34, 2.53)	γss	2.57	(2.40, 2.74)	
measles → pertussis	βmp	0.11	(0.04, 0.18)	γmp	0.12	(0.00, 0.24)	
smallpox → pertussis	βsp	0.04	(−0.04, 0.12)	γsp	0.11	(−0.02, 0.23)	
pertussis → measles	βpm	0.13	(0.06, 0.20)	γpm	0.11	(−0.01, 0.23)	
smallpox → measles	βsm	0.15	(0.05, 0.24)	γsm	0.24	(0.08, 0.39)	
pertussis → smallpox	βps	0.05	(−0.03, 0.13)	γps	0.19	(0.05, 0.33)	
measles → smallpox	βms	0.21	(0.12, 0.31)	γms	0.38	(0.21, 0.54)	

The negative binomial model can be formally written as (3) yi,td∼NBexpηi,td,expαϕd+ϕid,

where the mean parameter ηi,td is defined as in Eq. (2), and the nationwide dispersion parameters αϕd and the local dispersion parameters ϕid depend on the response disease. The priors are the same as with the Bernoulli model, with the addition of αϕd ∼ N(0, 12), ϕd∼N0,σϕd2, and σϕd ∼ Gamma(2, 1). From the negative binomial model, we could then compute our quantity of interest, the probability of observing at least one death in a specific town and month, which could be compared with the corresponding estimates of the Bernoulli model.

As a scoring rule for the model comparison, we used the expected log predictive density (ELPD), which measures the goodness of the entire predictive distribution (Vehtari, Gelman & Gabry, 2017). The ELPD was estimated via an approximate leave-one-out cross-validation using the R package loo (Vehtari et al., 2023). We left out one month and town from all the diseases at a time to estimate the ELPD. Models with higher values of ELPD correspond to greater posterior predictive accuracy for predicting new data points compared to models with lower ELPD values.

According to the differences in the ELPDs in Table 2, the best performing model is the one utilizing Bernoulli distribution and considering the diseases dependent. Omitting the dependencies results in the second-best model, with the difference in ELPD over three times the standard error. As could be expected, omitting the seasonal effect further impairs the model. When it comes to the negative binomial models with counts as responses, the order of the dependent, independent, and seasonless models is the same. The Bernoulli models outperform the negative binomial ones in all cases. Overall, we see that directly using the dichotomized data versus modeling the count data has a greater impact than considering the infection dependencies or seasonality in our model. However, even though in terms of predictive performance the differences between different Bernoulli models are relatively small, we used the most complex model in our main analysis. This is in line with the common Bayesian paradigm of incorporating the uncertainty of the model structure in the model (Vehtari & Ojanen, 2012).

Table 2 Differences of the ELPDs and the standard errors of the ELPD differences for the leave-one-out cross-validation.

The values in the first two columns are computed over all the years for the models estimated with the full data, and the last two columns are the values calculated over the last two years for the models discarding those years while estimating the models.

	Full data	Last two years	
	ELPDdiff	SEdiff	ELPDdiff	SEdiff	
Bernoulli dependent	0.00	0.00	0.00	0.00	
Bernoulli independent	−36.71	11.75	−46.84	4.21	
Bernoulli dependent, without season	−80.50	13.08	–	–	
Negative binomial dependent	−353.88	43.75	−381.38	16.70	
Negative binomial independent	−400.59	45.01	−490.50	18.90	
Negative binomial dependent, without season	−437.72	45.98	–	–	

For the dependent and independent models, we performed additional prediction checks by discarding the last two years of the data and estimating the probabilities for those years. We also calculated the ELPDs considering the removed years, see Table 2. The modifications without the seasonal effect were not included in this comparison due to their already evident poor performance and the fact that they were originally fitted merely to investigate the importance of the obvious seasonal variation. The posterior means were quite similar in all cases, but the posterior intervals were wider for measles and smallpox in the case of the negative binomial model, as can be seen from Fig. 7.

Figure 7 The dark gray lines depict the proportions of the towns where at least one death was observed.

The turquoise lines show the posterior means of the corresponding estimates and the shaded areas their 95% posterior intervals in the case of the Bernoulli model, whereas the pink lines and areas represent the same values for the negative binomial model. The dotted vertical line indicates the time point after which the model estimates are predicted by the models estimated without the data of the last two years.

Overall, the results of the Bernoulli, as well as the negative binomial, models seem to indicate similar interdependencies between the diseases. In the case of the negative binomial model, the estimates of all the time and town invariant parameters with their prior and posterior intervals are shown in Supplementary Table 2 on GitHub. Also, figures corresponding to the ones representing the results of the Bernoulli model are available in GitHub (Supplementary Figures 3-9).

Furthermore, to inspect the effect of the earlier disease history, a Bernoulli model according to Eqs. (1) and (2) was fitted with additional lags of two months for the three focal and three neighbor covariates. The approach resulted in having six scalar regression coefficients (three βs and three γs) more for each response disease than in our main model. The results are well aligned with the one-month lag model. The amount of available observations decreases by about three percentage points when introducing the two-month lag since we must know not only the previous observation but also the one preceding that. Thus this model is not completely comparable to our main model. Using the same data for both one-month and two-month lag models, the model with two-month lag performs better, measured with the ELPD: one-month lag model results in ELPDdiff =  − 989.87 and SEdiff = 53.89, compared with the two-month lag model. Nevertheless, there are some convergence and efficacy issues. Out of the 20, 000 iterations, there are 28 diverging ones which can potentially bias the results, thus they are not completely reliable (Betancourt, 2018). The outperformance of the two-month lag model in the sense of the ELPD might be also related to smaller variance regardless of the possible bias, which is consistent with the seemingly better fitting predictions gained from our main model than the two-month lag model (see Supplementary Figure 10). The full results are in GitHub in Supplementary Table 3 and Supplementary Figures 10-16.

Additionally, we fitted a model using our data aggregated on a weekly level, which increases the amount of missing data from 24% in the monthly data to 48% in the weekly data. We used lags from 1 to 4 weeks to cover as much delayed effect as with our main model. Due to the unequal number of weeks per calendar month, incorporating the monthly effect is not straightforward, so we omitted the seasonal effect. Unfortunately, this model did not converge, potentially because of the increased amount of missing data, or the complex dependency structures due to varying sub-monthly incubation and time-to-fatality times. The weekly data and model code are available in the supplementary material.

Discussion

We developed a Bayesian model to explore the spatio-temporal dynamics and co-dynamics of three fatal childhood infections—measles, smallpox, and pertussis—in pre-healthcare Finland (1820–1850). The main novelties of the approach are, firstly, the consideration of both the spatial and temporal aspects simultaneously, and, secondly, considering the connections not only within but also between the three diseases. Furthermore, our dataset is substantially different in comparison to the corresponding previous epidemiological literature. Instead of data regarding large cities or being pooled over countries, we exploited records from a sparsely populated nation, where 1.2–1.6 million inhabitants were spread over vast areas in hundreds of small towns without modern healthcare. Our model allows the inclusion of several explanatory elements which all capture different features. According to our results, all the components are meaningful and statistically distinctive, and the incorporation of the possibility of dependencies between the diseases leads to a model describing the data better than one merely assuming independent diseases. The data and the model framework are available on GitHub, providing a template for other researchers.

Based on our results, the main components explaining the temporal and geographical variation in the probabilities of observing at least one death caused by pertussis, measles, or smallpox are the nationwide incidence factors with their local adjustments. The estimated incidence factors follow the temporal behavior of the observed data, and the regional adjustments resemble the spatial patterns of the data (Figs. 2 and 4, and 3 and 6).

Measured by pairwise correlations of the incidence factors, a distinctive positive co-occurrence of measles and pertussis was discovered. Previous research has found positive, negative, and inconsistent co-occurrences of these infections, see, e.g., Rohani et al. (2003), Coleman (2015), and Noori & Rohani (2019). We also found a notable connection between measles and smallpox with a model without the seasonal component, but this correlation is not present in the full model including the seasonality. This indicates that their dynamics follow a similar, seasonal pace. Overall, the seasonal effect is visible among all the diseases. In addition to the nationwide incidence level, the seasonality increases the mortality during the first half of the year, depending on the disease, see Fig. 5. The seasonalities may reflect increased transmission during social gatherings, or they can be due to some environmental and climatic drivers (Metcalf et al., 2009; Metcalf et al., 2017). The work of Briga et al. (2021), based on selected data covering longer periods, indicates that of the infections of pertussis, measles, and smallpox only pertussis was linked with new year and Easter in Finland in the 18th and 19th centuries.

Furthermore, lagged dependencies within and between the infections were discovered as positive temporal and spatial effects of the explanatory variables. Recorded deaths in the focal town and in its adjacent towns in the previous month increased the risk of dying of the same disease. Between the infections, these effects were notably smaller (Table 1). It should be noticed that the coefficients reflecting the effect of the history of the focal town and its neighbors are not directly comparable, as the value of the focal covariate is either 0 or 1, but the neighborhood covariate is a proportion between 0 and 1.

According to the results, the risk of succumbing to pertussis, measles, and smallpox was mediated by occurrences of the other infections in the area. All these three diseases tended to increase the mortality related to the two other diseases, as all the pairwise interaction parameter estimates are positive. This might be due to general immunosuppression or to decreased condition following the previous infection. The strongest associations were found between measles and pertussis, and measles and smallpox. The possibility that pertussis is driven by the immunosuppressive effects of measles as suggested by Coleman (2015) and Noori & Rohani (2019) implies that the risk of dying of pertussis is increased by a recent measles infection. This is also supported by findings of Mina et al. (2015) showing that measles vaccination, by preventing measles-associated immune memory loss, decreases the risk of other infections. Our observations (see Table 1) are aligned with these results. However, also a reverse connection was recovered: the recorded deaths caused by pertussis in the same town during the previous month increased the risk of observing one or more measles induced deaths almost equally. A stronger lagged effect was discovered between measles and smallpox. Also these interactions were found to act in both directions.

To gain further insights into the specific effects of immunosuppression and impaired health conditions, longer than the one-month (or two-month) lags that we used here, should likely be employed. Unfortunately, our data do not suffice for identifying such effects as accounting longer histories or using finer timescale is challenging due to the missing data and the relative rarity of the deaths. Also herd immunity would be an important aspect to consider, but until proper population size estimates are available, it remains a topic for future work. Concerning our results, the lack of controlling for the longer term immunity might obscure some of the findings when compared to more contemporary datasets. Although immunosuppressive mechanisms of measles are well known in the literature, for many other diseases those are less known, for example, the effect of pertussis on measles (however, see, e.g., Macina & Evans (2021)). Thus, we suggest carefulness in interpreting our results as they might reflect the shorter term effects caused by the overall condition of the patients rather than true immunosuppression.

The observed spatially varying local risks of at least one death due to pertussis, measles, or smallpox may arise from the closeness of potential sources of infection, differences in cultural, housing, or nutritional circumstances, or even genetics (Honkola et al., 2018; Voutilainen, 2017; Kerminen et al., 2017). As can be seen from Fig. 3, the probabilities of detecting one or more deaths caused by pertussis and smallpox were greater in the eastern parts of Finland, whereas measles was clearly an infection emphasized in the southern parts, being in concordance what was suggested by Pitkänen, Mielke & Jorde (1989) and Ketola et al. (2021).

When it comes to the long term temporal behavior of the infections, it seems that epidemics in small populations, consisting of sparse metapopulations of tiny towns, might be dominated by reintroductions and fade-outs rather than by endemic dynamics more typical in densely populated cities and countries (Keeling & Grenfell, 1997; Grenfell & Harwood, 1997; Rohani et al., 2003; Ketola et al., 2021). In Briga, Ketola & Lummaa (2022) epidemics were found to reoccur in cycles of roughly four years in the 18th and 19th centuries in chosen Finnish towns with the highest quality data. The length and phase of such patterns are likely to vary due to annual and geographical differences in seasons, making them challenging to estimate from our scarce data. Our study covering 31 years did not reveal any long-term nationwide periodicities.

We modeled the deaths caused by measles, smallpox, and pertussis via a binary Bernoulli distribution, where value 1 denotes that there was at least one reported death given the disease, town, and month, and 0 for no reported deaths. This approach, while sacrificing some detail, allowed us to capture the broad trends and patterns in the data, and to make meaningful inferences about the spatio-temporal co-dynamics of these diseases. In contrast to the generally held view that dichotomizing data should be avoided, in our case directly modeling binary presence-absence data seemed to be beneficial compared to modeling observed death counts, potentially due to accuracy issues in the actual counts. However, both approaches led to practically identical main conclusions. The model comparisons also exemplified how our approach is applicable to other kinds of responses than Bernoulli variables.

We accounted for spatial dependencies using explanatory variables based on a neighborhood structure defined by a shared border between two towns. To model and quantify the evident epidemiological transmission dynamics, we included neighbor effects enabling the situation in the adjacent towns in the previous month to affect the probability to observe one or more deaths in the focal town. Our choice of neighborhood is straightforward, omitting the actual intensity of communication between the neighboring towns, hence possibly shrinking or magnifying the true dynamics of the infections. If there were more detailed data or complementary information about the social connections, other definitions for neighborhood, even with an appropriate weighing mechanism, could be employed. We tried to consider each pair of neighbors individually, but the information in the data was not sufficient for model identifiability, owing to the rarity of cases in neighboring towns. Naturally, including alternative appropriate and available covariates as explanatory variables is possible as well. The general spatio-temporal model developed for the purpose of exploring the dynamics and co-dynamics particularly in the case of sparse and scarce data is applicable to other corresponding datasets, for example, based on the historical parish records from other Nordic countries, or data on modern day rural areas.

The authors wish to acknowledge CSC –IT Center for Science, Finland, for computational resources. The authors also thank Virpi Lummaa.

Additional Information and Declarations

Competing Interests

Author Contributions

Data Availability

1 This text was originally published as a preprint (https://export.arxiv.org/abs/2310.06538).

The authors declare there are no competing interests.

Tiia-Maria Pasanen conceived and designed the experiments, performed the experiments, analyzed the data, prepared figures and/or tables, authored or reviewed drafts of the article, and approved the final draft.

Jouni Helske conceived and designed the experiments, performed the experiments, analyzed the data, authored or reviewed drafts of the article, and approved the final draft.

Harri Högmander conceived and designed the experiments, performed the experiments, authored or reviewed drafts of the article, and approved the final draft.

Tarmo Ketola conceived and designed the experiments, performed the experiments, analyzed the data, authored or reviewed drafts of the article, and approved the final draft.

The following information was supplied regarding data availability:

The data, code files for analyses, and additional figures and tables are available at GitHub and Zenodo:

- https://github.com/tihepasa/infectionDynamics/releases/tag/v2.0.2

- tihepasa. (2024). tihepasa/infectionDynamics: infectionDynamics supplements (v2.0.1). Zenodo. https://doi.org/10.5281/zenodo.11183838

10.5281/zenodo.10591760.

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
