# Peer review of "Spatio-temporal modeling of co-dynamics of smallpox, measles, and pertussis in pre-healthcare Finland"

_PeerJ, doi:10.7717/peerj.18155_

## Round 0.1 · original submission · Major Revisions

Thank you for your interesting submission. All three reviewers commented positively on your manuscript and commended the clear writing, motivation, and model design, but also make detailed suggestions to help improve the manuscript. Please review these carefully and consider how they can be incorporated.

In particular, please pay attention to:

1. Clarifying choices made about how data is used (ie, aggregation, handling missing data, etc) and include sensitivity analyses where appropriate to evaluate the impact of assumptions.

2. Ensure that the interpretation of the model outcome is correct, clearly described, and - where possible - validated and/or subject to appropriate sensitivity analyses.

3. Reviewer 3 suggests that your statistical model would be complemented by a simulation model that enabled deeper investigation of interactions between pathogens. I encourage you to consider this, but recognise that it may constitute a substantial piece of work beyond the scope of the current study. At a minimum, please consider the points raised, appropriately modify claims made on the basis of the statistical model results, and discuss how these may be addressed - and the findings strengthened - in future work.

Reviewer 1 ·

Basic reporting

Thank you for the opportunity to review “Spatio-temporal modelling of co-dynamics of smallpox, measles and pertussis in pre-healthcare Finland”, which aims to present a model to describe disease co-dynamics suitable for settings with scarce and sparse data. The manuscript was interesting to read and the idea of being able to use older data to develop methods for modelling infectious diseases is appealing.

The manuscript is clearly written, uses appropriate scientific language and there are only a few cases where the meaning could be clearer. The context of the study is well described, and the gap – that co-dynamics of infectious diseases are not often examined outside of large population centres – is motivated by the literature review.

Actions required:
1) Lines 34–35: Highlighting that understanding the relationships between zika, dengue and chikungunya is a serious challenge is not relevant for this paper and should be removed.
2) Lines 53–54: The population is described as ‘geographically isolated’ and so there is a need to describe why it is assumed that they are epidemiologically linked, especially given the age of the data and limited travel at that time.
3) Advantage of the model: While the gap is clearly articulated, it would be useful to describe how filling this gap will be an advance, given that the data are old and so some of the assumptions made may no longer be relevant. Is the method likely to translate well to contemporary data, particularly where there may be much greater deviation from dichotomy?

Experimental design

The authors use a Bayesian spatio-temporal model to describe disease (smallpox, measles, pertussis) co-dynamics in Finland in the 1800s, when towns varied in size from 300 to 25000. The Bernoulli model is appropriate given the probability outcome, and there appear to be adequate data to estimate the number of parameters being fitted. There are a few areas, listed below, where further justification of the assumptions would generate more confidence in the model outcomes. A major consideration is that the authors need to rethink what the outcome of their model represents, because it is not the probability of death as they state.

4) The model outcome stated on line 63 (and lines 104–105) is not correct: Due to the dichotomous nature of the model, the model does not estimate the death probabilities, but rather the probability that at least one death occurred in that region in that month. For example, in figure 1, the peak for smallpox at ~0.6 clearly shows that the probability of death and the proportion of towns where a death occurred do not mean the same thing. Calculation of the probability of death requires that the number of deaths and population denominator are considered. Please ensure that the interpretation of results throughout the manuscript reflects the correct meaning.

5) Use of presence/absence only: While the authors state that most monthly counts are still only 0 or 1 even after aggregation, it would strengthen belief in the model results if some data were available to reassure the reader that this is the case. Given the large variation in population size (300 to 25000), it is possible there could be a large variation in incidence. The ‘raw data’ provided appear to only include the presence/absence data (i.e. the spreadsheet states the data are dichotomous). Please provide either the case counts per month per region, or at least the distribution of case counts per month, so the reader can be satisfied that it is reasonable to assume that the data are dichotomous.

6) The missing data need further discussion: Given that the model extrapolates to regions with no data available, the reader needs to be sure that the missingness is not related to the death rate in that region. For example, could regions be unpopulated, completely isolated from other populations etc. Although the authors state at lines 90–91 that the deficiencies are not necessarily independent of the true number of deaths, some narrative on what is known about the reasons for missingness would again increase confidence in the model.

7) Assumed value for focal region when missing: Perhaps I missed it, but where data were missing for the focal region, what value was assumed for the focal region (i.e. the ‘x’s when calculating the focal region, and the ‘z’s when the missing focal region was a neighbour) in the model? Please add the methodology for the missing regions to the model section.

8) Predictions for missing regions: Following on from points 6 and 7 regarding missing data, there seems to be a pattern in the predictions for the regions with missing data. Based on Figure 2 (by eye), it seems as though most regions with missing data fall into the top quartile for pertussis and the second highest quartile for measles and smallpox. It would be worth examining the quartiles each of the missing regions falls into, to ensure that the algorithm is not biased, which may be related to the assumptions in comment 7).

9) Figure 2: While it is useful comparing maps of the data and the modelled output, the quartiles are a rather crude way to compare these results. It would be very informative to assess model accuracy to provide a similar map showing the quartiles of the differences between the model and the data for each region.
10) Line 119: Please define ‘nationwide incidence’. Given the description of the data, it is not clear whether it is the monthly nationwide incidence of death for a given disease, or the proportion of regions that had a least one death in the month (or something else).

Validity of the findings

Changes noted in the sections above need to be made in order to assess validity of the findings.

·

Basic reporting

English is professional and the paper is well written. The context and goal of the paper are clear. Good visualization and discussion of data.

The references to the statistics literature is good (I can't comment as much on references to the epidemiology literature). One minor comment: since the inception of NUTS, the default sampler in Stan has received several improvements discussed in Betancourt (2018): "A Conceptual Introduction to HMC".

Experimental design

Research question is well-defined and relevant to PeerJ. The data is clearly presented and so is the model used to analyze the data (this is further helped by the fact that the authors provide the Stan code used for their analysis).

I have a few questions/comments:
- Did the authors check, for instance when doing posterior predictive checks, that the posterior uncertainty was well-calibrated? For example, in Fig. 1 for smallpox, it looks like most of the data is within the 95% posterior interval, but I expect some (roughly 5%) to be outside. It almost looks like the posterior intervals are too large. Do the authors have an explanation for this?
- How was the quality of the model's prediction assessed? It seems the authors check whether the model fitted the data well, but it would be good to check predictions on out-of-sample data and forward in time.
- The discussion of the posterior intervals for the parameters is clear and insightful.
- I appreciate that the authors made model comparisons to understand the role of the interdependency parameters and the ELPD is a reasonable quantity to examine. However, I find ELPD difficult to interpret: how meaningful is a difference of 36.1 between two models? Can the author provide ways to interpret this quantity? It could be good to complement ELPD with a more interpretable (if less sophisticated) prediction score, and with some figures which plot the predictions made by different models.

Validity of the findings

As far as I can tell, the data and code to reproduce the results are provided on GitHub.

The conclusions are well stated, answer the research question and supported by the results.

Additional comments

Overall, I think this is an engaging paper and it requires some revision before being published in PeerJ.

I'll emphasize that my primary area of expertise lies in statistics. The epidemiological content of the paper is clearly explained and looks sound to me, though I cannot comment how it fits within the broader epidemiology literature.

·

Basic reporting

The article is well-written and clearly structured.

Experimental design

To strengthen the results, the proposed model should be validated using a simulation study based on transmission models with interaction mechanisms. For more specific comments, see my detailed review.

Validity of the findings

The validity of the findings (especially those related to the putative interactions between measles, pertussis, and smallpox) is currently unclear. See my detailed review for suggested improvements.

Additional comments

As explained in my detailed review, I would be happy to help the authors design the simulation to validate their regression model.

---

## Round 0.2 · Minor Revisions

Thank you for your consideration of the reviewers' comments. While reviewers were generally appreciative of the revisions made in response, they have highlighted several remaining issues to be addressed to ensure that results are clearly communicated and appropriately interpreted.

Please pay particular attention to:

1. Clarifying / justifying values plotted in Figure 2 (Reviewer 2).

2. Potential sensitivity analyses on key model choices (perhaps included as supplementary information) that can illustrate how model outputs depend on these choices (Reviewer 3, comments 2, 4).

3. Ensuring model interpretation and conclusions are well grounded in model outputs (Reviewer 3, comments 3, 6).

4. Careful proofreading of revised text (Reviewer 1).

Reviewer 1 ·

Basic reporting

The additional information provided in the manuscript has helped clarify the issues raised in the first review. The figure demonstrating the distribution of the data is a useful addition, and changing the maps to a continuous distribution rather than the use of quartiles has greatly improved their interpretation. All captions contain sufficient information to enable understanding of figures and tables.

Experimental design

The modelled outcome for the Bernoulli model has now been correctly reported throughout the manuscript. The negative binomial model results added to the manuscript, and their comparison to the Bernoulli model results provide a more convincing argument that the use of dichotomous data is appropriate for these data and this research question. The out of sample analysis supports the robustness of the model.

Validity of the findings

As far as I can tell, the code and data that enable the results to be reproduced all appear to have been provided on Github. Conclusions are supported by the results presented.

Additional comments

I appreciate that you have taken my feedback on board and thank you for comprehensively addressing my comments. I am satisfied with all the responses. Do check the newly added text carefully as some typographical errors have crept in.

·

Basic reporting

Same comment as in first review.

Experimental design

Same comment as in first review.

Validity of the findings

Same comment as in first review.

Additional comments

The authors have addressed most of my comments. I appreciate the extension of the section on Model Comparison.

I'm still not convinced by Figure 2 (formally Figure 1). It took me a while to understand the authors' explanation and I didn't find much clarification either in the text or in the figure's caption.

I believe it should be possible to plot data generated by the model (i.e. draws from p(y'|y)) and the observed data y, as done in posterior predictive checks. That way the lines would be "fully compatible". Right now the comparison is unclear and the uncertainty calibration cannot be checked.

I understand the value of plotting "predictions", but as the authors explain in their rebuttal, what they show are not predictions but "fitted values" (I'm not convinced this is the right term to use, that said, the formula for the posterior distribution clarifies what the authors mean---however this expression is missing from the revised manuscript and only appears in the response). A comment on notation: for the first term, use p(y'_t | y_1, ...) to distinguish the variable y'_t being simulated from the observed y_t.

In any case, I don't understand the value of plotting p(y'_t | y_1, ..., y_{t-1}, theta)p(theta | y_1, ..., y_T). Either plot predictions (as done in Figure 7) or plot retrodictions, i.e. p(y'_t | y_1, ..., y_T).

I recommend the authors either (a) plot p('_t | y_1, ..., y_T) or (b) clarify in the caption in the text what they are plotting and provide justification for plotting "fitted values".

If the authors do this, I recommend accepting the paper and I do not need to see manuscript again.

·

Basic reporting

The article is well-written and clearly structured.

Experimental design

Still unclear—see specific comments below.

Validity of the findings

Still unclear—see specific comments below.

Additional comments

I thank the authors for their responses. Although some of my comments have been addressed, the authors' responses are, on the whole, underwhelming. The authors seem to have just dismissed several of my major comments, which I considered critical issues. I understand the data limitations, but this should be no excuse for being so dismissive and doing nothing. The interpretation of these results—particularly the positive correlations between pathogens—is still unclear. I go back to my major comments below.

1) Given the editor’s decision, I understand the choice not to run this simulation study. However, I disagree that such a study would not be feasible or fruitful. As I wrote in my initial review, there is now relatively extensive evidence from such studies showing that purely statistical methods (like the one proposed by the authors) are ineffective at inferring pathogen interactions. These shortcomings probably stem from the inability of these methods to capture the latent process, particularly population immunity.

2) Although the new count model is interesting, I would have expected the authors to test at least another level of temporal aggregation. So the question remains: why monthly, and how do the results depend on this arbitrary choice? Also, the authors’ argument that binarizing the data reduces the uncertainty is statistically dubious and likely illusory.

3) OK, but see my comment above—population immunity is not just an element of discussion but a critical part of any model pretending to capture the transmission dynamics of infectious diseases.

4) It didn’t seem so difficult to test at least another time lag. Again, this shouldn’t be just a discussion point, given the claim (unfounded, in my opinion) that the positive correlations could indicate immunosuppressive effects.

5)Given that the mean age of infection/death for pertussis seems significantly lower than that for measles and smallpox (and the fact that pertussis does not induce a systemic infection), an interaction pertussis–>smallpox or pertussis–>measles now seems even less plausible.

6) The fact that infections co-occur does not tell much, per se, about pathogen interactions. See also my comment above about pertussis—I don’t know any biological evidence for pertussis causing immunosuppression. The interpretation of the positive correlations is still unclear (in fact, it could just reflect confounding bias, as the authors now acknowledge), and I wouldn’t even list potential biological mechanisms in the abstract.

---

## Round 0.3 · accepted · Accept

Thanks you for your further response and edits. I have assessed this revision and am happy that the current version of the manuscript is read for publication.